# Adapting Rewards to the Agent Using Rational Activation Functions

## Abstract

Fixed environment rewards can lead to miscalibrated gradients, instability, and inefficient learning when signals are poorly scaled relative to the agent's updates. We introduce **Rational Reward Shaping (RRS)**, a reward transformation that converts raw rewards into normalized signals aligned with the agent's experience. RRS combines experience-normalized scaling with a monotone rational activation to reshape sensitivity and curvature while preserving reward order. It adapts automatically to changing reward regimes and integrates seamlessly into standard actor–critic updates–simply replacing the immediate reward in the target–requiring minimal code changes and no task-specific reward engineering. Across DDPG, TD3, and SAC on six MuJoCo benchmarks, RRS consistently improves average returns in both noiseless and perturbed-reward settings, with larger gains under noise, while incurring only 6% average wall-clock overhead. RRS provides a general, plug-and-play method to produce better-calibrated reward signals, strengthening learning without modifying environment design. Source code is available at: `https://github.com/anonymouszxcv16/RRS`

## 1 Introduction

Deep reinforcement learning (DRL) faces the persistent challenge of *exploration–exploitation trade-off* Sutton & Barto (2018). On one hand, *exploration* focuses on investigating new states and actions in order to expand the agent's knowledge (Dabney et al., 2020; Russo et al., 2018; Sekar et al., 2020). On other hand, *exploitation* emphasizes the use of the agent's current knowledge to select actions that maximize immediate rewards (Pomerleau, 1989; Fujimoto & Gu, 2021; Fujimoto et al., 2019; Haarnoja et al., 2018; Chen et al., 2021). Although exploration and exploitation pursue seemingly opposing goals, they ultimately share the same objective: maximize long-term cumulative reward. Successfully navigating this trade-off requires a guiding mechanism that fosters cooperation between the two goals over time, leading to effective decision making Konda & Tsitsiklis (1999).

A critical aspect of this trade-off lies in the agent's **experiences**. Each agent collects and reuses a distinct set of experiences through its replay buffer, which directly shapes its learning dynamics and policy formation. These stored experiences determine how effectively the agent can transform environmental signals into actionable decisions Lin (1992). Imbalance in replayed experiences can restrict generalization and adaptability, whereas excessive or poorly structured replay can increase computational cost and reduce sample efficiency without meaningful learning gains.

Against this backdrop, reward shaping has emerged as a widely adopted strategy to balance exploration and exploitation by guiding the learning process toward more effective behaviors Ng et al. (1999a). By modifying the reward signal, reward shaping enhances performance in sparse-reward environments. However, environment-provided rewards are typically fixed and uninformed by an agent's experiential characteristics. This misalignment interacts directly with the **diversity and quality** of the agent's collected experiences. Agents trained on *low-quality but high-diversity* data (e.g., from broad D4RL datasets Fu et al. (2020)) are prone to bias and instability due to the presence of out-of-distribution (OOD) actions Fujimoto & Gu (2021), while agents trained on *high-quality but low-diversity* data risk overfitting to narrow behavioral patterns and failing to generalize effectively.

In this study, we propose **Rational Reward Shaping (RRS)**, a method that transforms external rewards into an *internal reward* signal aligned with the agent's experiences. RRS applies a monotonic

transformation to raw rewards using rational activation functions Delfosse et al. (2021a;b) and normalization of stored experiences, reshaping the reward curvature and gradient behavior according to the agent's experience distribution and task structure. This enables the agent to exploit memory more effectively, refine internal representations, and improve both sample efficiency and computational utilization. RRS is also *adaptive*: the shaping parameter is adjusted online based on reward variability, progressively flattening the transformation as the reward distribution stabilizes to emphasize finer reward differences while limiting bias.

We evaluated RRS on continuous control benchmarks using the DDPG, TD3 and SAC algorithms. Our results demonstrate consistent improvements across all configurations in noiseless and noisy environments, with the latter showing a larger gain for RRS. These improvements are consistent across all evaluated algorithms, indicating that RRS is generic and robust. Our contributions are twofold:

1. *Rational Reward Shaping (RRS)*: we present the RRS algorithm along with empirical validation on standard MuJoCo continuous control tasks, demonstrating its effectiveness through experience-based performance improvements.

2. *Open-source implementation*: we provide publicly available RRS source code to facilitate transparency, reproducibility, and ease of integration into existing reinforcement learning pipelines.

## 2 RELATED WORK

Reward shaping Ng et al. (1999b); Zou et al. (2019); Ng et al. (1999a); Serban et al. (2017) has long been used to adjust reward signals for faster learning and more stable convergence. It provides a principled way of injecting domain knowledge into reinforcement learning systems, which can lower sample complexity when done without altering the underlying objective Gupta et al. (2022). A central approach is potential-based reward shaping Grzes & Kudenko (2010), which directs exploration by adding differences of potentials across state transitions while still preserving the optimal policy Wiewiora (2003). Classic work shows that potential-based shaping is equivalent to certain Q-value initializations, highlighting how it speeds up early training by biasing initial estimates toward more informative parts of the state space Wiewiora (2003). More recent surveys situate shaping within broader design strategies, combining it with exploration bonuses, intrinsic motivation, and preference-based signals. These works stress approaches that avoid specification gaming while still maintaining asymptotic correctness Ibrahim et al. (2024).

Extensions of shaping target challenges like sparse-reward domains and complex credit assignment Andrychowicz et al. (2017b); Sutton (1984). Some methods use structured schedules or exploration-guided shaping, where intrinsic signals are learned to guide behavior when external rewards are delayed or noisy Devidze et al. (2022). Analytical studies also help clarify when and why engineered rewards improve learning efficiency–typically by reducing exploration demands and identifying favorable conditions Gupta et al. (2022). Another related direction is human-in-the-loop shaping, where evaluative feedback is integrated directly as reward signals. Systems like TAMER and Deep TAMER treat human input as shaping signals, enabling interactive reward design that can outperform demonstration-only strategies in high-dimensional problems Warnell et al. (2018).

Beyond making rewards denser, researchers have tackled temporal credit assignment by redistributing returns rather than adding auxiliary shaping terms Pignatelli et al. (2023). RUDDER is one key method, decomposing returns to shift delayed signals back to earlier decisions that caused them, thereby aligning immediate feedback with causal actions and improving long-horizon training Arjona-Medina et al. (2019). Temporal Value Transport follows a similar logic, but uses attention mechanisms over episodic memory to propagate value to distant, causally relevant events, again without changing the task objective Hung et al. (2019). Another structural approach is Reward Machines, which encode automata-like structures over rewards. This representation supports automated shaping, decomposition, and counterfactual relabeling, strengthening performance in both single-task and multi-task settings Icarte et al. (2018).

While effective, most reward shaping and redistribution methods rely on predefined strategies that do not adapt to task difficulty. An alternative approach focuses on curriculum design and task selection, where adjusting training difficulty can accelerate learning without costly manual reward

engineering Portelas et al. (2020); Bengio et al. (2009). Empirical results in automatic curriculum learning indicate that strategies requiring little or no expert knowledge can be competitive, offering a practical option when explicit reward design is expensive or risky Romac et al. (2021).

Another refinement is reward shuttering, where shaped signals are gradually reduced or gated during training. This approach captures the early efficiency benefits of shaping while ensuring that the agent ultimately relies on the true objective Ng et al. (1999a). Conceptually, it ties back to the policy-invariance guarantees of potential-based shaping and analyses that attribute efficiency gains to early-stage value biases Wiewiora (2003); Hu et al. (2020). In practice, shuttering typically involves scaling down shaping coefficients or potentials over time, often based on measures like progress, uncertainty, or performance thresholds. It can also be combined with dynamic shaping for robustness when agent representations shift during training Devlin & Kudenko (2012). Recent work with language-model-assisted pipelines suggests that text-to-reward techniques can support shuttering by permitting automated, auditable, and progressively refined shaping as the agent's competence improves Xie et al. (2023).

**Novelty.** Our work advances reward shaping by introducing a dynamically adaptive mechanism that directly links reward design to the agent's experiences. Unlike prior approaches, we integrate rational activation over normalized rewards with an auto-tuned curvature parameter $\alpha$, which adapts online to the reward distribution's variance. This design ensures sensitivity under noisy conditions and sharper guidance as convergence improves.

## 3 METHODOLOGY

The **Rational Reward Shaping (RRS)** framework extends the Actor-Critic family of algorithms Konda & Tsitsiklis (1999); Lillicrap et al. (2015); Fujimoto et al. (2018); Haarnoja et al. (2018) and is compatible with continuous control environments (Brockman, 2016; Todorov et al., 2012; Fu et al., 2020) in both online and offline settings. RRS aims to transform the environment's raw reward signal into a form better aligned with the agent's experience distribution, facilitating more effective learning and generalization. The approach is grounded in two established principles: (1) **reward normalization**, which has been shown to improve stability and credit assignment in RL (Naik et al., 2024), and (2) **rational activation functions**, which enhance gradient flow and learning efficiency in deep RL (Delfosse et al., 2021a). Accordingly, RRS consists of two main components: **Normalization** and **Activation**.

### 3.1 NORMALIZATION

The rewards provided by the environment can vary wildly in scale and are often unbounded. This makes learning more difficult, especially for agents with limited memory. We aim to address this by normalizing the reward signal using statistics from the agent's replay buffer $\mathcal{D}$. Given a raw reward $r$, we normalize it as follows:

$$\overline{r} = \frac{r - \mathbb{E}_{r' \in \mathcal{D}}(r')}{\max_{r' \in \mathcal{D}}(r')} \tag{1}$$

where $\mathbb{E}_{r' \in \mathcal{D}_t}(r')$ denotes the **current** mean experience replay rewards and $\max_{r' \in \mathcal{D}}(r')$ denotes the **current** the maximum experience replay rewards.

This transformation ensures that the rewards the agent receives remain within a a bounded range, making it easier for the agent to interpret them. Intuitively, our normalization is similar to computing an advantage function, at the level of the reward.

### 3.2 RATIONAL ACTIVATION

Following the reward normalization, we apply a non-linear transformation by using a rational activation function Delfosse et al. (2021a). This step adds useful curvature to the reward, which we hypothesize will enable the agent to learn more reliably:

$$r^{\text{shaped}} = \frac{1}{\alpha} \cdot \frac{1}{1 + e^{-\alpha \bar{r}}} \cdot \log\left(1 + e^{\alpha \bar{r}}\right) \tag{2}$$

This function is smooth, always positive, and controlled by a single parameter $\alpha$. Lower $\alpha$ results in more gradual shaping, while higher values create sharper distinctions between high and low rewards. This shaped reward preserves each experience value with regard to its neighbors, but adjusts its scale and curvature to better match the agent's experiences. Moreover, since the RRS transformation is both **positive** and **monotonic** (being the composition of two monotonic functions), it **preserves policy optimality** in accordance with the theoretical framework established by Ng et al. (1999b).

### 3.3 Adaptive Tuning of the Rational Function

While fixing $\alpha$ in Eq. 2 already improves the performance our DRL baselines (see Section 5), keeping this hyperparameter static can be unstable across different environments. To address this, we enable the agent to adapt $\alpha$ automatically. In high-variance reward regimes, RRS will tend to select smaller $\alpha$ to stabilize learning and remain sensitive to fine differences, whereas low-variance settings will lead to larger $\alpha$ to accentuate informative signals and yield more decisive gradients for policy improvement. The auto-tuning mechanism uses observed reward statistics to steer $\alpha$ accordingly, requiring no manual intervention.

### Window-Based Update Rule

We use the inverse of the reward signal's standard deviation to guide $\alpha$ selection. Intuitively, a high reward standard deviation is more distinguishable on the flat region of the RRS function, favoring a lower $\alpha$, while a low standard deviation is better distinguished on the steep region, favoring a higher $\alpha$. To stabilize updates, we employ a window-based averaging mechanism controlled by a user-defined parameter $w \in [1, T]$ ($T$ is the terminal step), which accumulates the inverse reward dispersion over time and updates $\alpha$ periodically. At each step $t$, we define the instantaneous inverse reward dispersion:

$$\xi_t = \frac{1}{\text{std}_{r \in \mathcal{D}_t}(r)} \tag{3}$$

We accumulate this value over a moving window of size $w$:

$$\Xi_t = \sum_{i=t-w+1}^{t} \xi_i \tag{4}$$

Then, at every window interval, we update the shaping parameter using the average inverse standard deviation:

$$\alpha \leftarrow \text{scaled\_sigmoid}(\frac{1}{w} \cdot \Xi_t) \tag{5}$$

where:

$$\text{scaled\_sigmoid}(x) = \alpha_{\min} + \frac{\alpha_{\max} - \alpha_{\min}}{1 + e^{-x}} \tag{6}$$

After the update, the accumulator $\Xi_t$ is reset. This adaptive approach is effective because it enables RRS to adapt both to the environment and to changes in the DRL agent's exploration preferences.

### 3.4 Integration into Critic Update

Our proposed reward shaping approach can be easily integrated in any DRL algorithm with a critic update step. All that is required is to replace the original reward $r_t$ with our reshaped reward $r_t^{\text{shaped}}$:

Table 1: The properties of the evaluated MuJoCo tasks.

| Task | $\dim(S)$ | $\dim(A)$ | $\frac{\mathrm{std}(R)}{|\mathrm{mean}(R)|}$ |
|------|-----------|-----------|------------------|
| **Continuous** | | | |
| Hopper | 11 | 3 | 0.63 |
| Cheetah | 17 | 6 | 2.28 |
| Walker | 17 | 6 | 1.18 |
| Ant | 27 | 8 | 1.92 |
| Humanoid | 376 | 17 | 0.04 |
| Standup | 376 | 17 | 0.16 |

**Task Difficulty Interpretation**: Cheetah and Ant exhibit higher reward variability and lower coordination demands, making them comparatively easier locomotion tasks. Hopper and Walker require more structured balance, reflected in lower reward dispersion. Humanoid and Standup are the most challenging: both operate in a high-dimensional action–state space with dense reward signals (low normalized std), requiring fine-grained stability and coordinated control.

$$y_t = r_t^{\mathrm{shaped}} + \gamma Q_{\mathrm{target}}(s_{t+1}, a_{t+1}) \tag{7}$$

where $Q_{\mathrm{target}}$ is the output of target critic network.

The simplicity by which RRS can be integrated into existing algorithms is a significant advantage, as it requires minimal adaptation. Moreover, the inclusion of RRS does not prevent the use of other methods that are designed to boost the performance of DRL algorithms such as Hindsight Experience Replay Andrychowicz et al. (2017a) or Intrinsic Curiosity Module Pathak et al. (2017).

## 4 EXPERIMENTAL SETUP

**Baselines.** We evaluate three variants of our proposed RRS approach. The first is RRS(auto), which uses the automatic tuning of the $\alpha$ parameter. The two other variants are RRS(0.5) and RRS(1), that use fixed $\alpha$ values of $0.5$ and $1$, respectively.

We integrate our approach into three widely-used DRL algorithms: Deep Deterministic Policy Gradient (DDPG) Lillicrap et al. (2015), Twin Delayed DDPG (TD3) Fujimoto et al. (2018), and Soft Actor-Critic (SAC) (Haarnoja et al., 2018). For DDPG and TD3, we additionally apply Reward Centering (RC) Naik et al. (2024), implemented on top of the single-critic (DDPG) and double-critic (TD3) actor-critic architectures.

We compare the performance of the RRS-enhanced version of each baseline to its original version.

**Evaluation metric.** Performance is measured using the standard **average cumulative reward** metric in continuous-control RL (Lillicrap et al., 2015; Fujimoto et al., 2018; Haarnoja et al., 2018). For each episode, cumulative reward is computed as the sum of all rewards from the initial state until termination, and the final score is obtained by averaging across evaluation episodes to reduce variance and improve statistical stability. Statistical significance follows conventional notation: [*] for ($p < 0.05$), [**] for ($p < 0.01$), and [***] for ($p < 0.001$). For experiments involving noisy rewards, improvements are computed relative to each algorithm's corresponding **non-noise baseline** to ensure consistent comparison across evaluation settings.

**Evaluation environments.** We evaluated RRS across six standard MuJoCo continuous-control tasks (Table 1). All algorithms were trained for 1 million time-steps and repeated across five fixed random seeds for reproducibility and fair comparison. The normalized reward standard deviation reported in the table was estimated using a fully random, non-learning DDPG rollout over 1 million time-steps sampled from the final replay buffer distribution. All experiments were executed on a GPU cluster equipped with NVIDIA RTX 6000 Ada-Generation hardware.

**Noiseless and noisy rewards.** We use our evaluation environments in two settings: *noiseless* and *noisy*. In the noiseless (normal) setting, the agent observes all reward signals correctly. In the noisy setup, the agent receives a perturbed reward signal that affects the latter's perception of its performance. We model the noisy environment as a perturbed-reward MDP $\tilde{\mathcal{M}} = (\mathcal{S}, \mathcal{A}, R, C, P, \gamma)$,

Table 2: The results of our proposed approach under the *non-noisy* reward setting. The reported improvement (%) represents the average of the per-environment gains, computed as: $\frac{\text{competitor}_{\max}}{\text{baseline}_{\max}} - 1$. where the baseline corresponds to the standard version of each algorithm without RRS.

| Environment | DDPG | RC | RRS (0.5) | RRS (1) | RRS (Auto) |
|---|---|---|---|---|---|
| Hopper | $2,144 \pm 1,092$ | $1,296 \pm 505$ | $3,187 \pm 406$ | $\mathbf{3,378 \pm 276}$ | $3,020 \pm 831$ |
| Cheetah | $12,512 \pm 390$ | $12,442 \pm 785$ | $12,389 \pm 212$ | $\mathbf{12,961 \pm 331}$ | $12,751 \pm 283$ |
| Walker | $\mathbf{3,221 \pm 1,894}$ | $3,136 \pm 1,057$ | $1,721 \pm 843$ | $2,521 \pm 1,160$ | $1,553 \pm 1,199$ |
| Ant | $1,796 \pm 1,683$ | $1,679 \pm 1,771$ | $\mathbf{4,274 \pm 1,130}$ | $3,121 \pm 1,242$ | $3,143 \pm 1,791$ |
| Humanoid | $820 \pm 370$ | $\mathbf{2,003 \pm 1,252}$ | $772 \pm 569$ | $1,040 \pm 699$ | $844 \pm 442$ |
| Standup | $178,559 \pm 58,188$ | $124,978 \pm 66,409$ | $\mathbf{184,474 \pm 42,421}$ | $154,741 \pm 32,990$ | $172,709 \pm 76,766$ |
| Improvement | – | $10.8\%^{*}$ | $\mathbf{22.7\%}$ | $21.1\%^{*}$ | $10.9\%^{*}$ |
| **Environment** | **TD3** | **RC** | **RRS (0.5)** | **RRS (1)** | **RRS (Auto)** |
| Hopper | $\mathbf{3,580 \pm 70}$ | $1,096 \pm 78$ | $3,358 \pm 140$ | $3,450 \pm 128$ | $3,352 \pm 142$ |
| Cheetah | $9,603 \pm 5,721$ | $10,882 \pm 3,377$ | $12,297 \pm 401$ | $\mathbf{13,323 \pm 116}$ | $12,255 \pm 385$ |
| Walker | $\mathbf{5,147 \pm 901}$ | $3,233 \pm 2,669$ | $2,014 \pm 485$ | $3,952 \pm 746$ | $4,237 \pm 835$ |
| Ant | $3,890 \pm 4,105$ | $6,499 \pm 652$ | $6,510 \pm 122$ | $\mathbf{6,936 \pm 74}$ | $6,450 \pm 381$ |
| Humanoid | $5,248 \pm 2,775$ | $\mathbf{5,869 \pm 5,256}$ | $3,398 \pm 3,034$ | $4,604 \pm 2,450$ | $5,652 \pm 233$ |
| Standup | $155,053 \pm 10,140$ | $157,777 \pm 62,497$ | $174,073 \pm 28,530$ | $\mathbf{215,734 \pm 51,695}$ | $180,013 \pm 22,043$ |
| Improvement | – | $-2.1\%$ | $0.9\%^{*}$ | $\mathbf{19.5\%}^{**}$ | $15.5\%^{***}$ |
| **Environment** | **SAC** | | **RRS (0.5)** | **RRS (1)** | **RRS (Auto)** |
| Hopper | $3,113 \pm 910$ | | $3,420 \pm 133$ | $3,384 \pm 36$ | $\mathbf{3,423 \pm 126}$ |
| Cheetah | $9,229 \pm 5,506$ | | $10,694 \pm 1,224$ | $\mathbf{11,866 \pm 1,310}$ | $11,653 \pm 1,266$ |
| Walker | $\mathbf{4,998 \pm 640}$ | | $2,979 \pm 1,450$ | $4,818 \pm 860$ | $4,844 \pm 76$ |
| Ant | $3,627 \pm 3,898$ | | $917 \pm 14$ | $\mathbf{5,469 \pm 1,459}$ | $4,768 \pm 2,505$ |
| Humanoid | $5,117 \pm 2,768$ | | $4,419 \pm 2,354$ | $\mathbf{5,553 \pm 168}$ | $5,477 \pm 303$ |
| Standup | $135,948 \pm 18,740$ | | $163,563 \pm 15,855$ | $158,272 \pm 4,360$ | $\mathbf{200,669 \pm 39,711}$ |
| Improvement | – | | $-13.8\%^{***}$ | $18.2\%^{**}$ | $\mathbf{19.9\%}^{**}$ |

where, at each time step $t$, the true reward $r_t \in R$ is not observed directly Wang et al. (2020). The agent receives a perturbed version of the reward $\tilde{r}_t \in \tilde{R}$, generated by the function $C : \mathcal{S} \times R \to \tilde{R}$.

In our experiments, we focus on the state-independent corruption case, where the noisy reward is obtained via multiplicative perturbations of the form:

$$\tilde{r} = r + \eta, \quad \text{where} \quad \eta = \text{sign} \cdot r \cdot \xi \cdot \beta$$
$$\xi \sim \mathcal{U}(0,1), \quad \text{sign} \in \{-1, +1\} \tag{8}$$

Here, $\beta \in [0,1]$ is the maximum noise fraction (`noise_frac_max`), and $\xi$ is a uniformly sampled scalar controlling the perturbation magnitude. The random sign produces unbiased noise, ensuring that the expected corruption is zero. This formulation naturally extends to continuous-reward environments and can simulate diverse real-world conditions where sensor readings, feedback loops, or evaluative signals are inconsistent or unreliable. In all noisy setting experiments, we used $\beta = 0.01$.

The goal of the noisy setting is to evaluate the DRL algorithms in a more challenging and realistic scenario. Perturbed rewards are common in the real world: inaccurate sensor readings, delayed transmissions, accidental mouse clicks, etc. This setup stresses the importance of robustness in DRL algorithms.

## 5 RESULTS

**Noiseless setting.** Table 2 presents results under noiseless rewards. For each baseline (DDPG, TD3, SAC), we compare the standard implementation to three RRS variants: RRS(0.5), RRS(1), and RRS(auto). Across all algorithms, RRS(1) and RRS(auto) consistently achieve the strongest gains, with overall improvements of $18.2\%$–$21.1\%$ and $10.9\%$–$19.9\%$, respectively. These improvements are statistically significant: DDPG with RRS(1) gains $+21\%$ ($p < 0.05$), TD3 with RRS(auto) $+15\%$ ($p < 0.001$), and SAC with RRS(auto) $+20\%$ ($p < 0.01$). As hypothesized in Section 3.3, in low-variance reward settings, stronger reward curvature (larger $\alpha$) enhances signal separability and accelerates policy learning, while lower $\alpha$ may under-differentiate rewards, reducing efficiency.

Table 3: The results of our proposed approach under noisy reward conditions. The reported improvement (%) is computed as the average of the per-environment gains, defined by: $\frac{\text{competitor}_{\max}}{\text{baseline}_{\max}} - 1$, where the baseline corresponds to the same algorithm trained in the *non-noisy* setting.

| Environment | DDPG | RRS (0.5) | RRS (1) | RRS (auto) |
|---|---|---|---|---|
| Hopper | $21 \pm 11$ | $2,690 \pm 785$ | $2,580 \pm 765$ | $\mathbf{3,373 \pm 271}$ |
| Cheetah | $-18 \pm 6$ | $\mathbf{13,040 \pm 318}$ | $12,073 \pm 380$ | $12,534 \pm 250$ |
| Walker | $2 \pm 11$ | $\mathbf{2,321 \pm 800}$ | $1,425 \pm 580$ | $1,787 \pm 990$ |
| Ant | $965 \pm 17$ | $3,043 \pm 2,807$ | $\mathbf{4,101 \pm 1,095}$ | $3,821 \pm 1,559$ |
| Humanoid | $193 \pm 135$ | $891 \pm 359$ | $\mathbf{968 \pm 405}$ | $944 \pm 645$ |
| Standup | $66,331 \pm 13,804$ | $224,847 \pm 52,278$ | $\mathbf{227,355 \pm 19,174}$ | $220,324 \pm 29,961$ |
| Improvement | -80.8% | 22.5% | 17.6% | **27.4%** |
| **Environment** | **TD3** | **RRS (.5)** | **RRS (1)** | **RRS (Auto)** |
| Hopper | $3,375 \pm 135$ | $3,365 \pm 1,461$ | $3,399 \pm 95$ | $\mathbf{3,406 \pm 85}$ |
| Cheetah | $\mathbf{12,732 \pm 736}$ | $11,916 \pm 2,470$ | $12,657 \pm 325$ | $12,403.0 \pm 496.7$ |
| Walker | $\mathbf{4,520 \pm 2,476}$ | $2,975 \pm 1,448$ | $4,218 \pm 1,248$ | $4,157 \pm 977$ |
| Ant | $5,305 \pm 2,275$ | $6,324 \pm 1,947$ | $\mathbf{6,777 \pm 174}$ | $6,694 \pm 197$ |
| Humanoid | $\mathbf{6,488 \pm 330}$ | $4,462 \pm 2,534$ | $5,605 \pm 103$ | $4,437 \pm 2,386$ |
| Standup | $148,427 \pm 40,057$ | $200,252 \pm 70,016$ | $172,770 \pm 19,452$ | $\mathbf{214,702 \pm 48,414}$ |
| Improvement | 11.7% | 8.8% | **19.5%** | 16.7% |
| **Environment** | **SAC** | **RRS (.5)** | **RRS (1)** | **RRS (Auto)** |
| Hopper | $3,115 \pm 860$ | $2,870 \pm 1,757$ | $3,384 \pm 36$ | $\mathbf{3,486 \pm 198}$ |
| Cheetah | $\mathbf{12,457 \pm 2,209}$ | $11,158 \pm 4,523$ | $11,866 \pm 1,310$ | $11,423 \pm 433$ |
| Walker | $\mathbf{5,584 \pm 598}$ | $1,919 \pm 1,145$ | $4,818 \pm 860$ | $3,631 \pm 2,615$ |
| Ant | $3,386 \pm 2,142$ | $916 \pm 793$ | $\mathbf{5,469 \pm 1,459}$ | $3,631 \pm 2,615$ |
| Humanoid | $4,798 \pm 2,196$ | $4,399 \pm 2,464$ | $\mathbf{5,553 \pm 168}$ | $4,499 \pm 2,372$ |
| Standup | $149,151.5 \pm 13,651$ | $\mathbf{160,617 \pm 47,130}$ | $158,272 \pm 4,360$ | $126,001 \pm 71,957$ |
| Improvement | 7.3% | -19.9% | **18.2%** | -0.1% |

Notably, applying Reward Centering (RC) yields mixed results. RC performs best on Humanoid–the task with the most complex reward structure, lowest normalized reward standard deviation, and largest state and action spaces–achieving roughly +10% over standard DDPG. However, RC wins only 1 out of 6 tasks overall and even slightly underperforms on TD3 (-2.1%), indicating that while it can help in very complex environments, its benefits are limited compared to RRS, which provides more consistent improvements across tasks.

**Noisy setting.** Table 3 reports results when stochastic noise is injected into the reward signal (Equation 8), with improvements measured relative to each baseline's noiseless performance. TD3 and SAC are relatively robust to noise, sometimes even exceeding noiseless performance (+11.7% and +7.3%, respectively), whereas DDPG suffers substantial degradation. Importantly, RRS mitigates noise effects: for DDPG, RRS(0.5) maintains its improvement, RRS(1) drops slightly to +17.6%, and RRS(auto) nearly *triples* its gain to +27.4%, highlighting the effectiveness of adaptive $\alpha$ in unstable regimes. Even for robust learners like TD3 and SAC, RRS(1) and RRS(auto) provide additional gains.

Specifically, when noise is applied to the reward signal, setting $\alpha = 1$ results in an average performance gain of 18%, whereas $\alpha = 0.5$ yields only a 4% improvement under the same conditions. This further highlights that stronger or adaptively tuned reward shaping is crucial for maintaining learning efficiency in stochastic environments.

**Summary.** Across both noiseless and noisy rewards, RRS consistently improves performance for all evaluated DRL baselines. Notably, the adaptive $\alpha$ variant delivers the most reliable enhancement across tasks, achieving an average gain of 15% compared to 11% for fixed $\alpha$, demonstrating that adaptive tuning consistently boosts learning, rather than serving as an environment-specific heuristic. While lower shaping strength ($\alpha = 0.5$) may occasionally be optimal for sensitive algorithms like DDPG, robust learners generally benefit most from higher or adaptively tuned $\alpha$.

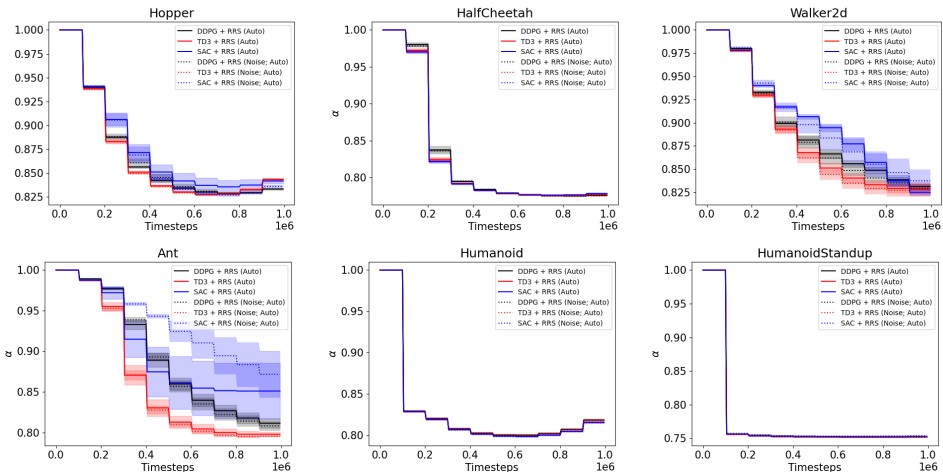

Figure 1: The changes in the $\alpha$ value of RSS(auto) throughout its training.

Table 4: Wall-clock runtime comparison (in minutes) between DDPG, TD3, and SAC, and their RRS-based variants. Results are reported as the mean $\pm$ standard deviation over five random seeds.

| Environment | DDPG | RRS (Auto) | TD3 | RRS (Auto) | SAC | RRS (Auto) |
|---|---|---|---|---|---|---|
| Hopper | **143.1 $\pm$ 5.1** | 154.3 $\pm$ 49.1 | **170.3 $\pm$ 9.3** | 185.5 $\pm$ 53.4 | **132.4 $\pm$ 3.4** | 164.6 $\pm$ 3.3 |
| Cheetah | 152.9 $\pm$ 19.0 | **142.5 $\pm$ 3.4** | 176.5 $\pm$ 7.2 | **162.1 $\pm$ 3.3** | **145.8 $\pm$ 3.3** | 162.1 $\pm$ 3.9 |
| Walker | **138.2 $\pm$ 7.0** | 154.3 $\pm$ 52.0 | 178.3 $\pm$ 2.1 | **160.7 $\pm$ 4.8** | **141.3 $\pm$ 3.7** | 161.0 $\pm$ 4.1 |
| Ant | **149.8 $\pm$ 6.9** | 173.0 $\pm$ 67.1 | 179.8 $\pm$ 10.8 | **172.0 $\pm$ 2.1** | **146.6 $\pm$ 2.5** | 172.7 $\pm$ 2.8 |
| Humanoid | **155.7 $\pm$ 5.5** | 177.2 $\pm$ 51.9 | **200.2 $\pm$ 12.0** | 222.9 $\pm$ 46.8 | **165.0 $\pm$ 0.6** | 185.5 $\pm$ 0.6 |
| Standup | 219.3 $\pm$ 3.5 | **192.1 $\pm$ 7.7** | 243.1 $\pm$ 5.7 | **210.5 $\pm$ 2.4** | **192.8 $\pm$ 2.6** | 211.8 $\pm$ 1.3 |
| Improvement | – | 4.9% | – | **-2.6%** | – | 14.9% |

# 6 ANALYSIS & DISCUSSION

## 6.1 ANALYZING THE AUTOMATIC TUNING PROCESS

To better understand the behavior of RRS(auto), we tracked the evolution of its $\alpha$ values during training, shown in Figure 1. We set the hyperparameters $\alpha_{\min} = 0.5$ and $\alpha_{\max} = 1$, based on their practical effectiveness in our experiments. Across all tasks, $\alpha$ gradually decreases over time, reflecting adaptation to the growing reward standard deviation as the agent explores more complex states.

Furthermore, in all tasks except Ant–a simpler environment with the second-highest normalized reward standard deviation–the $\alpha$ values in noisy conditions are equal to or lower than in the noiseless setting. This is expected: adding noise increases the reward standard deviation, making the environment effectively more complex, and our adaptive scheme assigns lower $\alpha$ values in response. This behavior aligns with our design (see Sections 3.2 and 3.3), confirming that higher reward variance in more challenging conditions leads to appropriately reduced shaping strength.

## 6.2 ANALYZING THE REWARD DISTRIBUTION

We now analyze the consistency of the obtained rewards throughout the baselines' training. For each 5,000 training steps, we calculated the standard deviation of the obtained rewards. We then plotted these values for each algorithms' entire training process. The results, presented in Figure 2, show that for every baseline, its RSS(auto) version has lower standard deviation, than the original algorithm. These results support our hypothesis that our proposed approach stabilizes the data collection policy.

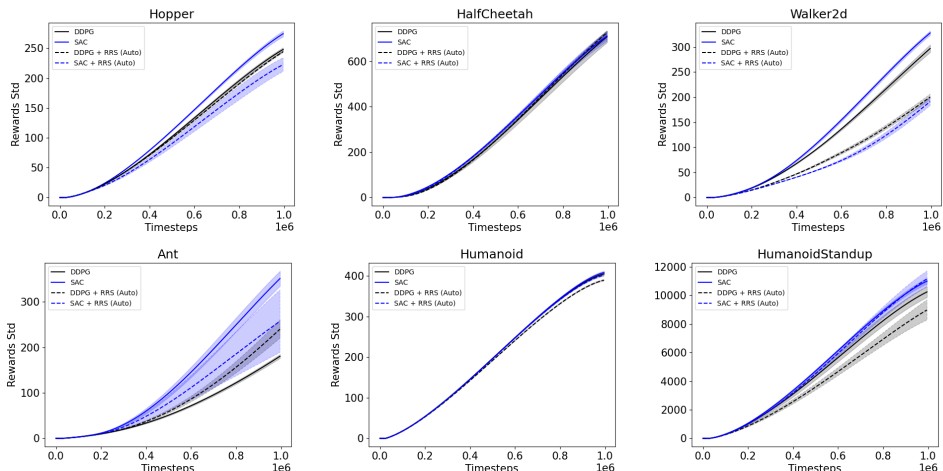

Figure 2: The temporal evolution of reward standard deviation for the three evaluated DRL algorithms and their corresponding RRS-augmented variants.

### 6.3 RUNTIME ANALYSIS

Table 4 presents a wall-clock runtime comparison between the baseline DRL algorithms (DDPG, TD3, SAC) and their RRS-augmented counterparts. The results show that RRS introduces only a minor computational overhead in most cases. Specifically, for TD3, the RRS (Auto) variant achieves a modest **2.6% reduction** in average runtime, demonstrating that the adaptive reward shaping can even streamline learning dynamics. For DDPG, RRS incurs only a **small overhead** of approximately 4.9%, reflecting efficient integration with minimal additional cost. Although SAC with RRS shows a **runtime increase** of roughly 14.9%, this overhead is justified by the substantial performance gains observed in the corresponding learning curves, highlighting a favorable trade-off between efficiency and effectiveness.

## 7 CONCLUSION

Rational Reward Shaping (RRS) aligns environment rewards with agent experiences by normalizing replay-based rewards and applying a monotonic rational activation with adaptively tuned curvature. Across six MuJoCo tasks and three actor-critic backbones, RRS consistently improves performance, achieving an average gain of 15% on DDPG and TD3 (including $\alpha = 0.5$, $\alpha = 1$, and $\alpha_{\text{auto}}$) under noiseless conditions–substantially higher than Reward Centering (RC), which yields only +4% on average.

These benefits come with minimal cost, incurring only 6% overall wall-clock overhead, and integrate seamlessly into existing DRL algorithms.

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

## A    APPENDIX

## LLM USAGE STATEMENT

In accordance with the policy on large language model (LLM) usage, we acknowledge that LLMs were used as general-purpose assistive tools in this research. Specifically:

1. We used **ChatGPT** and **Perplexity AI** to identify and surface relevant literature during the preparation of the related work section. These tools were employed to perform preliminary web-based searches and generate concise summaries to aid our understanding of recent and historical research.

2. We also used both tools to help reformulate and improve the clarity of individual sentences across multiple sections of the paper. The core ideas, structure, and technical content were developed independently by the authors.

LLMs were not used to generate or fabricate any experimental data, analysis, or novel technical content. All outputs from the models were critically reviewed and verified by the authors. We take full responsibility for the content and claims made in this paper.

**LLM Tools Cited:**

1. ChatGPT Achiam et al. (2023)
2. Perplexity AI Team (2025)

