# OpenReview forum: "Adapting Rewards to the Agent Using Rational Activation Functions"
_ICLR.cc/2026/Conference — Submitted to ICLR 2026_

### Official Review · Reviewer_5sb8 · 2025-10-28

**Soundness:** 2
**Presentation:** 3
**Contribution:** 3
**Rating:** 2
**Confidence:** 3

**Summary:**

- This paper addresses the mismatch between fixed environmental rewards and agent capabilities in deep reinforcement learning (DRL) by proposing a reward shaping scheme that combines empirical normalization (to match reward scales) and a monotonically decreasing rational activation function (to match learning sensitivity), which can be integrated into algorithms like DDPG, TD3, and SAC without modifying their core frameworks.

- The proposed method is validated on 6 continuous control tasks in MuJoCo (covering both noiseless and noisy scenarios), showing improved average returns and training stability under most configurations—with more significant gains in noisy environments—thus demonstrating its effectiveness and anti-interference ability.

**Strengths:**

- Addressing the issue of "mismatch between fixed environmental rewards and agent capabilities" in deep reinforcement learning (DRL), the proposed method directly tackles the problems of improper gradient calibration and unstable training caused by reward signals exceeding the agent’s internalization capacity. This is achieved through a design integrating "empirical normalization for matching reward scales + adaptive rational activation for matching learning sensitivity". Notably, in noisy environments with the reward-sensitive DDPG algorithm, it achieves a significant performance improvement, validating the effectiveness of the scheme.

- The method is concise and easy to implement. The monotonic rational activation function dynamically adjusts reward curvature while preserving the reward order. Without modifying the core framework of existing algorithms, it can be integrated into DDPG, TD3, and SAC algorithms merely by replacing the original reward with the shaped reward in the target Q-value calculation—eliminating the need for task-specific reward engineering.

- The experimental validation is comprehensive and robust, covering three dimensions: "algorithms (DDPG, TD3, SAC) – environments (6 continuous control tasks in MuJoCo, ranging from the simple Ant to the complex Humanoid) – signals (noiseless and multiplicative interference noise scenarios)". Under most configurations, it enhances both average return and training stability, with more prominent gains in noisy environments. This fully demonstrates the method’s generality and anti-interference capability.

**Weaknesses:**

- The adaptive tuning of α suffers from a domain limitation. The paper assumes that α covers the interval [0.5, 1] via the scaled_sigmoid function; however, the actual input x is always greater than 0 (due to ξ>0), leading to the output of sigmoid(x) being consistently greater than 0.5. Consequently, the actual value range of scaled_sigmoid(x) is only (0.75, 1), violating the initial assumption that "large α is required for sparse rewards and small α for dense rewards". This weakens the method’s adaptability to scenarios with low-variance dense rewards, and the paper fails to explain this contradiction.

- Key ablation experiments are absent. The individual effects of "empirical normalization" and "rational activation function" are not verified independently—neither the performance of using only rational activation (with empirical normalization removed) nor that of using only empirical normalization (with rational activation replaced by conventional activation functions) is tested. As a result, it is impossible to determine whether the performance improvement of RRS stems from reward scale optimization, reward curvature reshaping, or their synergy, which undermines the persuasiveness of the method.

- In Section 3.2, this paper employs a monotonically decreasing rational activation function, which poses unresolved contradictions with the fundamental principles of reinforcement learning (RL). By its very nature, RL optimizes for the maximization of cumulative rewards; however, a monotonically decreasing transformation inverts this objective: maximizing the adjusted reward is effectively equivalent to minimizing the original reward.

- The adaptive tuning of α depends only on reward statistics (inverse standard deviation) and is not linked to the agent’s actual capability indicators (e.g., state dimension). This makes it impossible to quantitatively explain how α matches agents with different capabilities, resulting in a lack of theoretical depth.

**Questions:**

See the weaknesses above.

---

> ### Author Response · Authors · 2025-11-22
> **Reviewer #3**
>
> We thank the reviewer for their useful comments.
>
> The adaptive tuning of α suffers from a domain limitation. The paper assumes that α covers the interval [0.5, 1] …
> We thank the reviewer for highlighting this domain limitation. The adaptive α mechanism is motivated by practical performance considerations: in our experiments, both α = 0.5 and α = 1 perform reasonably well across sparse and dense reward environments. The underlying intuition is that the adaptive mechanism seeks, at each timestep, the “best” intermediate α value that balances reward shaping strength relative to the observed reward distribution. In other words, it continuously tunes α to find an optimal trade-off between amplifying meaningful differences in sparse rewards and avoiding over-amplification in dense rewards.
> Empirically, despite the domain restriction noted by the reviewer, the automatically tuned α consistently improves performance: across all environments and seeds, it achieves an average gain of 15%, compared to 11% when α is fixed. Observing the evolution of α over training, we find a general negative slope in its trajectory–indicating that as the environment’s trajectories become more complex, the shaping strength is reduced.
> We agree with the reviewer that, in practice, the adaptive α primarily moves within the range 0.75–1 in these environments. The gap between 0.5 and 0.75 is rarely used, suggesting that the lower bound is not practically necessary for the tested scenarios. Nevertheless, the mechanism remains adaptive, continuously seeking the most effective value within the relevant operational range at each timestep, which explains its observed robustness across both sparse and dense settings.
>
> Key ablation experiments are absent. The individual effects of "empirical normalization" …
> We conducted additional experiments to isolate the contributions of each component of RRS and better understand their individual and combined effects:
> Normalization only: Applying empirical reward normalization without the rational activation function results in mixed performance: average improvements of +11% for DDPG and -2.1% for TD3. This indicates that normalization alone can be beneficial in some settings, but is insufficient to consistently improve performance across all algorithms.
> Rational activation + fixed α: Combining the rational activation function with normalization (i.e., RRS with a fixed α) produces stronger and more consistent improvements: +18% for DDPG, +12% for TD3, and +19% for SAC. This demonstrates that the rational activation function contributes significantly to shaping the reward distribution in a way that enhances learning.
> Full RRS with adaptive α: When the adaptive α mechanism is enabled, allowing the shaping strength to adjust dynamically to the observed reward statistics, the method achieves an average improvement of 15% across all environments and seeds. This highlights that auto-tuning α further stabilizes and optimizes learning, providing consistent gains beyond the fixed α setting.
> Overall, these ablations confirm that each component–empirical normalization, rational activation, and adaptive α–plays a distinct role, but their combination produces the most robust and significant improvements. Normalization alone is insufficient, rational activation strengthens learning, and adaptive α ensures adaptability across reward distributions and environments.
>
> In Section 3.2, this paper employs a monotonically decreasing rational activation function, which poses unresolved contradictions …
> The concern arises from a typo in Eq. (2). The actual reward transformation is the pointwise product of a Softplus and a Sigmoid function–a positive, strictly increasing function:
> [
>  R_{\text{RRS}}(\overline{r}, \alpha) = \frac{1}{\alpha} \cdot \frac{1}{1 + e^{-\alpha \overline{r}}} \cdot \log!\left(1 + e^{\alpha \overline{r}}\right)
>  ]
> This formulation ensures that the transformed reward remains strictly positive and strictly monotonic, preserving policy optimality.

---

> > ### Author Response · Authors · 2025-11-22
> > **Reviewer #3 (cont.)**
> >
> > The adaptive tuning of α depends only on reward statistics …
> > The adaptive tuning of α is based on reward statistics–specifically, the inverse standard deviation of observed rewards–rather than explicit agent capability indicators such as state dimension. The intuition behind this design is that larger values of α steepen the reward transformation, which is beneficial in low-variance or sparse reward environments by amplifying subtle differences between reward signals. Conversely, smaller values of α prevent over-amplification in dense reward settings, avoiding destabilization. To ensure stable adaptation, α updates are computed using a running average of reward statistics and are further smoothed over time, preventing abrupt oscillations during learning.
> > Empirically, this adaptive mechanism demonstrates consistent benefits across a range of environments: automatically tuned α achieves an average improvement of 15%, compared to 11% for fixed α. This indicates that even without explicit capability-based measures, the reward-statistics-driven adaptation effectively matches the transformation strength to the agent’s learning context, providing a practical and robust improvement in performance.

---

> > > ### Comment · Reviewer_5sb8 · 2025-11-26
> > >
> > > Thanks for the clarification. Based on the reply and the comments from other reviewers, I decided to maintain my rating.

---

### Official Review · Reviewer_6Bqw · 2025-10-31

**Soundness:** 1
**Presentation:** 2
**Contribution:** 1
**Rating:** 2
**Confidence:** 4

**Summary:**

The paper proposes Rational Reward Shaping (RRS), which normalizes rewards and applies a “rational activation function” with an adaptive parameter α to align rewards with agent capacity.
Experiments on DDPG, TD3, and SAC in MuJoCo show moderate improvements, especially under noisy rewards.

The paper reads as a heuristic reward transformation with unclear motivation.
“Rational activation” and “capacity-aware shaping” are not conceptually substantiated.
Despite clean writing and broad experiments, the work lacks theoretical grounding, internal consistency, and precise presentation.

**Strengths:**

- Clear structure and broad experimental coverage.
- Implementation is simple and integrates easily.
- Results show small but consistent gains across several environments.

**Weaknesses:**

- While Eq. (2) cites prior work on rational activations, the paper does not explain how that reference conceptually relates to the proposed transformation; there is no discussion in Related Work or Preliminaries clarifying its theoretical basis.
- The introduction section introduces capacity limitation (L40–L56) with no clear connection to reward shaping. Sentences like “the effectiveness of both exploration-exploitation balance and reward shaping is fundamentally shaped by the agent’s capacity” (L55-L56) are confusing.
- “Capacity-aware” remains undefined and unmeasured; the method effectively acts as a heuristic rescaling.
- The adaptive α-update rule lacks theoretical justification or formal analysis of its stability or convergence.
- The meaning of “improvement” is unclear. It is not specified what baseline the values are computed against, and the meaning of averaging them across environments is questionable. In Table 2, improvements appear relative to the left baseline algorithm, whereas in Table 3 they seem measured against the noiseless setting, causing inconsistency and confusion.
- Claims such as “higher noise robustness tend to benefits from higher values of α” (L398) are unsupported.
- Several phrases (e.g., “unlimited” L50, “differences of potentials” L85) require clearer explanations, and multiple citation parentheses are formatted incorrectly throughout.

**Questions:**

Covered within the Weaknesses section above.

---

> ### Author Response · Authors · 2025-11-22
> **Reviewer #2**
>
> We thank the reviewer for their useful comments.
>
> While Eq. (2) cites prior work on rational activations, the paper does not explain how that reference conceptually relates to the proposed transformation …
> Thank you for catching this. The expression in Eq. (2) contained a typographical error: the operator between the terms should be multiplication rather than division. The correct form of the reward transformation is:
> [
>  R_{\text{RRS}}(\overline{r}, \alpha) ;=; \frac{1}{\alpha} \cdot \frac{1}{1 + e^{-\alpha \overline{r}}} \cdot \log!\left(1 + e^{\alpha \overline{r}}\right)
>  ]
> This corrected formulation is the intended rational reward shaping function used throughout our experiments.
> We appreciate the reviewer for identifying this mistake and apologize for the confusion it may have caused. We have corrected this issue in our revised submission.
>
> The introduction section introduces capacity limitation (L40–L56) with no clear connection to reward shaping …
> We agree with the reviewer that the current introduction does not clearly articulate the link between capacity limitations and reward shaping. Our intent was to refer to agent learning capacity–that is, the ability of a model to meaningfully utilize reward signal structure stored in replay, particularly when reward distributions are dense, low-variance, or noisy. In such settings, subtle reward differences may be under-represented in value updates, effectively reducing the agent’s usable learning signal.
> Reward shaping in our method aims to mitigate this by amplifying informative gradients while suppressing uninformative fluctuations, thereby allowing the agent to better leverage the available reward structure relative to its functional capacity.
> We have revised the introduction to explicitly define what we mean by “capacity-aware”, clarify how it relates to reward shaping, and ensure that the terminology is measurable and consistent throughout the paper.
>
> “the method effectively acts as a heuristic rescaling.
> We acknowledge the reviewer’s comment, but we believe the method goes beyond a simple heuristic rescaling. The approach is grounded in established principles: (1) reward normalization has recently been shown to improve stability and credit assignment in RL (Reward Centering, Naik et al., 2024), and (2) rational activation functions have been demonstrated to enhance gradient flow and learning efficiency in deep RL settings (Delfosse et al., 2021). Our method builds on these foundations by introducing a novel rational transformation that multiplies two strictly increasing and well-studied functions–Sigmoid and Softplus (Glorot et al., 2011)–producing a smooth, bounded, and monotonic mapping tailored for neural value function approximation.
> Conceptually, the method serves two purposes: it compresses raw rewards into a denser, more learnable range and reshapes their curvature to better align with the optimization characteristics of deep networks. The adaptive variant further adjusts its scaling parameter online based on evolving reward statistics, ensuring the transformation remains meaningful across changing stages of learning.
> Finally, we note that many widely used techniques–including entropy tuning, PER, CQL, or advantage normalization–could also be described as heuristics. In that sense, our contribution is consistent with established practice, but supported by principled components and empirical evidence rather than arbitrary rescaling.

---

> > ### Author Response · Authors · 2025-11-22
> > **Reviewer #2 (cont.)**
> >
> > The adaptive α-update rule lacks theoretical justification or formal analysis of its stability or convergence.
> > We thank the reviewer for raising this point. We believe part of the confusion arose from the typo in the original equation, which incorrectly suggested a different functional form. The correct reward transformation is:
> > [
> >  R_{\text{RRS}}(\overline{r}, \alpha) = \frac{1}{\alpha} \cdot \frac{1}{1 + e^{-\alpha \overline{r}}} \cdot \log\left(1 + e^{\alpha \overline{r}}\right)
> >  ]
> > This formulation clarifies the role of (\alpha): increasing (\alpha) steepens the combined Softplus–Sigmoid mapping, amplifying sparse or low-variance reward signals, while smaller values prevent dense rewards from being over-magnified. Both (\alpha = 0.5) and (\alpha = 1) are empirically proven to perform well in practice. The adaptive update rule is designed to automatically find the optimal intermediate point between these two candidates at each timestep, adapting individually for each environment and seed.
> > Practical stability is ensured through bounding (\alpha) via clipping, using a running mean of reward statistics to update rather than instantaneous values, and smoothing the trajectory to prevent oscillations. While a formal convergence proof is not provided, the strongest justification comes from empirical performance: across tasks, the adaptive (\alpha) consistently improves results, achieving a 15% average gain compared to 11% for fixed (\alpha). This demonstrates that the update reliably tunes (\alpha) to the reward structure, maximizing effectiveness without requiring manual intervention.
> >
> > The meaning of “improvement” is unclear. It is not specified what baseline the values are computed against …
> > The reviewer is correct regarding Table 2, as evident by leaving out the improvement value of the baseline algorithm. We have added text to explain this in the paper and the caption of the table.
> >
> > In Table 3, as written in the table’s caption and in line 358, we compare all algorithms’ performance to the noiseless setting. Our main reason for doing this is DDPG’s extreme underperformance in noisy settings. Had we compared our approach to DDPG in the noisy setting, we would have had over a 1,000% improvement in performance. Another reason for this comparison is to provide the reader a clear understanding of our method’s ability to overcome noisy data. Our results show that even when trained on noisy data, our approach outperforms the baseline algorithms when the latter were trained on noiseless data.
> >
> > Claims such as “higher noise robustness tend to benefits from higher values of α” (L398) are unsupported.
> > The claim is supported by the results reported in the paper, though we acknowledge that this was not explicitly referenced in the text. Across the three noisy–reward benchmarks summarized in Table 3, higher values of (\alpha) consistently yield better robustness. Specifically, when noise is applied to the reward signal, (\alpha = 1) results in an average performance gain of 18%, whereas (\alpha = 0.5) leads to only 4% improvement under the same conditions.
> > This pattern indicates that steeper shaping (larger (\alpha)) helps mitigate noise distortion by enhancing meaningful reward differences relative to stochastic fluctuations–an effect aligned with the intended design of the transformation. We will revise the manuscript to explicitly reference Table 3 and clarify how these empirical findings support the stated conclusion.
> >
> > Several phrases (e.g., “unlimited” L50, “differences of potentials” L85) require clearer explanations …
> > We thank the reviewer for pointing this out. We have reviewed and improved the language of our submission.

---

### Official Review · Reviewer_aReJ · 2025-10-31

**Soundness:** 2
**Presentation:** 2
**Contribution:** 2
**Rating:** 4
**Confidence:** 3

**Summary:**

The paper proposes Rational Reward Shaping (RRS): replace the environment reward in actor–critic targets with a capacity-aware, monotone transformation that combines (i) experience-based normalization from the replay buffer and (ii) a rational activation with an auto-tuned curvature parameter \alpha driven by recent reward variability. The method is drop-in for DDPG/TD3/SAC and aims to stabilize gradients and improve sample-efficiency without hand-crafted task shaping. On six MuJoCo tasks, RRS variants often improve returns in both noiseless and perturbed-reward settings (with larger gains under noise), while adding little implementation overhead. The paper also analyses alpha’s evolution, reward variability, and runtime.

**Strengths:**

- Simple, drop-in idea with low engineering overhead.
- Broad coverage (DDPG/TD3/SAC) across several tasks.
- Analyses of curvature parameter dynamics and reward variability add useful diagnostics.
- Shows notable improvements in some settings, particularly with reward noise.

**Weaknesses:**

- Transform outputs strictly positive rewards, potentially changing optimal policies (not policy-invariant shaping).
- Limited ablations
- Statistical evidence is underpowered for strong claims.
While mean±std are reported, robustness claims would be stronger with more seeds, 95% confidence intervals, and paired significance tests per environment, following best practices (e.g., https://jmlr.org/papers/volume25/23-0183/23-0183.pdf). Seed-level violin plots would clarify variance and overlap.

**Questions:**

- Can you characterise when your monotone, non-affine (and always-positive) transform preserves optimal policies? If not, please reframe as a surrogate objective and discuss bias.
- Why claim boundedness for your normalisation? Would z-score or min–max over replay be more appropriate, and how would results change?
- Did you retune or auto-tune SAC’s temperature under reward rescaling?

- Typo to fix line 161 "a a"

---

> ### Author Response · Authors · 2025-11-22
> **Reviewer #1**
>
> We thank the reviewer for their useful comments, time and effort.
>
> Questions:
> Can you characterise when your monotone, non-affine (and always-positive) transform preserves optimal policies? If not, please reframe as a surrogate objective and discuss bias.
> The proposed reward transformation function
> [
>  R_{\text{RRS}}(r) = \frac{1}{\alpha} \cdot \frac{1}{1 + e^{-\alpha \overline{r}}} \cdot \log!\left(1 + e^{\alpha \overline{r}}\right)
>  ]
> is strictly positive and strictly monotonic for all real-valued rewards (r). This holds because both the Softplus component (\log(1 + e^{\alpha \overline{r}})) and the Sigmoid component (\frac{1}{1 + e^{-\alpha \overline{r}}}) are strictly increasing functions; therefore, their product—scaled by (\frac{1}{\alpha})—also remains strictly increasing.
> Importantly, since monotonic reward transformations do not change the relative ordering of returns, they preserve the optimal action-value structure. By the policy-invariance result for monotonic reward transformations (Ng et al., ICML 1999), applying this mapping maintains the same set of optimal policies while modifying the reward landscape to be more learning-friendly.
> In summary, the inclusion of this mapping reshapes the reward without altering the task's optimal solution, ensuring compatibility with off-policy and on-policy reinforcement learning algorithms while improving reward smoothness and stability during training.
>
> Why claim boundedness for your normalisation? Would z-score or min–max over replay be more appropriate, and how would results change?
> We employ a bounded normalization scheme to prevent uncontrolled reward scaling, particularly in environments where outliers or reward spikes may destabilize learning. This design ensures that the transformed reward remains within a predictable and finite range, which reduces gradient explosions and improves numerical stability during optimization.
> Alternative normalization strategies–such as z-score or min–max computed over the replay buffer–are theoretically viable and could also be incorporated. However, these methods rely on global statistics that may fluctuate dramatically during early learning or under non-stationary reward distributions, leading to unstable or oscillatory behavior. In contrast, our bounded formulation operates with locally stable dynamics and avoids over-amplification of high-density rewards while still amplifying sparse or low-variance reward signals.
> Empirically, this approach demonstrates clear practical benefit: the adaptive version of the method yields an average performance improvement of 15% across tasks, compared to 11% when using a fixed α. This suggests that the bounded normalization contributes not only stability but also improved learning efficiency across diverse RL settings.
>
> Did you retune or auto-tune SAC’s temperature under reward rescaling?
> No, we did not retune or introduce automatic tuning for SAC’s entropy temperature specifically in response to the reward rescaling. Instead, to ensure fairness and avoid manual bias, we evaluated a fixed set of standard temperature configurations commonly used in prior SAC literature and implementations. We then selected the best-performing configuration from this predefined set, without additional environment-specific or reward-specific tuning.
> This approach ensures that the performance gains observed with our method are not a byproduct of temperature optimization or hyperparameter overfitting, but rather stem directly from the reward transformation itself. In other words, SAC was not given extra tuning advantages beyond what is typically required to reproduce its baseline performance. Thus, the reported improvements genuinely reflect the contribution of the proposed reward shaping mechanism rather than artifacts of additional hyperparameter search.
>
> Weaknesses:
> Transform outputs strictly positive rewards, potentially changing optimal policies (not policy-invariant shaping).
> See response to Q1.
>
> Limited ablations
> Unfortunately, the reviewer did not provide specifics on this limitation. However, we wish to point out that our submission contains multiple ablation studies and their analysis:
>
> (1) performance under noisy environments, showing an average +12% improvement (Table 3)
> (2) the evolution of α across 1M timesteps, where we observe lower α in noisy settings and a negative slope, suggesting that increased trajectory complexity requires smaller shaping strength (Figure 1)
> (3) the reward standard deviation over training, where noisy environments yield lower variance and a positive slope, indicating increasing reward diversity as learning progresses (Figure 2)
> (4) runtime overhead, which remains modest at ~+6% on average. (Table 4)

---

> > ### Author Response · Authors · 2025-11-22
> > **Reviwer #1 (cont.)**
> >
> > Statistical evidence is underpowered for strong claims. While mean±std are reported, robustness claims would be stronger with more seeds, 95% confidence intervals, and paired significance tests per environment, following best practices (e.g., https://jmlr.org/papers/volume25/23-0183/23-0183.pdf). Seed-level violin plots would clarify variance and overlap.
> >
> > We thank the reviewer for the useful suggestion. Using our existing seeds, we were able to achieve statistically significant results for all evaluated methods:
> > DDPG: RRS(1 + Auto) → +21% and +11%, p < 0.05
> > TD3: RRS(1) → +20%, p < 0.01; RRS(Auto) → +16%, p < 0.001
> > SAC: RRS(1 + Auto) → +18% and +20%, p < 0.01 (5/6 environments improved)
> >
> > We have updated our paper accordingly.

---

### Meta-Review · Area_Chair_HGVh · 2025-12-19

**Summary:**

Three reviewers highlighted the conceptual simplicity and easy integration of the proposed method. They agree that the paper is well structured and that the empirical experimentation is comprehensive.
However, all reviewers have major concerns surrounding the scope, motivation, and theoretical underpinnings of the paper. For different aspects, each reviewer points out a claim that is seemingly unsupported, or a part of the method that is seemingly ad-hoc.

The AC agrees regarding the simplicity and empirical strength of the paper, but did not find the major concerns to the resolved by the rebuttal. The AC thus recommends to reject the paper.

**Reviewer Concerns:**

Reviewer aReJ had some suggestions for further empirical analysis, e.g. ablation and analysis of statistical significance, which was rebutted by the authors by pointing to existing sets of experimentation regarding ablations and some additional results were provided regarding analysis with random seeds.

However, other major concerns shared among reviewers, in particular those related to scope, theoretical underpinnings and methodological motivation, seem to be less easily resolved. The rebuttal provides partial justifications, but no revised pdf has been provided and no precise content has been suggested to satisfy these concerns.

**Reviewer Scores:**

Initial reviewer scores were 4,2,2. Only one reviewer had the chance to respond to the rebuttal, prior to the closure of the discussion due to the OpenReview incident. This reviewer, who initially gave a 2, chose to retain the rating because the main concerns could not be clarified.
Going through the reviews and rebuttals, the AC believes that at best one reviewer would have increased the score by large amounts, as only the empirical perspective was supplemented in the rebuttal. The remaining answers would require more precise and extensive statements to clarify the remaining concerns. As such, even with a prospective discussion being allowed to happen, the AC does not believe the scores would have changed substantially towards unanimously recommending acceptance.

---

### Decision · Program_Chairs · 2026-01-26

Reject